

# Pollen report: quantitative review of pollen crude protein concentrations offered by bee pollinated flowers in agricultural and non-agricultural landscapes

Tobias Pamminger, Roland Becker, Sophie Himmelreich, Christof W. Schneider and Matthias Bergtold

Global Ecotoxicology, BASF SE, Limburgerhof, Germany

## ABSTRACT

To ease nutritional stress on managed as well as native bee populations in agricultural habitats, agro-environmental protection schemes aim to provide alternative nutritional resources for bee populations during times of need. However, such efforts have so far focused on quantity (supply of flowering plants) and timing (flower-scarce periods) while ignoring the quality of the two main bee relevant flower-derived resources (pollen and nectar). As a first step to address this issue we have compiled one geographically explicit dataset focusing on pollen crude protein concentration, one measurement traditionally associated with pollen quality for bees. We attempt to provide a robust baseline for protein levels bees can collect in- (crop and weed species) and off-field (wild plants) in agricultural habitats around the globe. Using this dataset we identify crops which provide sub-optimal pollen resources in terms of crude protein concentration for bees and suggest potential plant genera that could serve as alternative resources for protein. This information could be used by scientists, regulators, bee keepers, NGOs and farmers to compare the pollen quality currently offered in alternative foraging habitats and identify opportunities to improve them. In the long run, we hope that additional markers of pollen quality will be added to the database in order to get a more complete picture of flower resources offered to bees and foster a data-informed discussion about pollinator conservation in modern agricultural landscapes.

## INTRODUCTION

The green agricultural revolution during the mid-20th century drastically increased productivity in agriculture and changed land use on a global scale (*Evenson & Gollin, 2003*). The combination of high-yield crop varieties, chemical fertilizer, plant protection products and intensified mechanization have amplified crop biomass production, which in turn has enabled the support of an ever-growing human population (*Evenson & Gollin, 2003*; *Pingali, 2012*). At the same time, the associated reduction in plant diversity in intensified agricultural habitats (e.g., large scale mass flowering crop-cultures) has been suggested to

Corresponding author
Tobias Pamminger,
tobias.pamminger@basf.com

adversely affect pollinator populations which provide essential ecosystem services (*Goulson et al., 2015*; *Potts et al., 2010*; *Roulston & Goodell, 2011*). In particular, bees (Hymenoptera: Apoidea), a diverse group of primarily phytophagous insects present in most major habitats around the globe (*Michener, 2000*; *Wcislo & Cane, 1996*), have recently come into focus because they provide a large share of pollination services in agricultural habitats and some populations are apparently in decline (*Goulson et al., 2015*). Bees rely solely on plant derived resources (pollen and nectar) to satisfy their nutritional needs (*Brodschneider & Crailsheim, 2010*; *Michener, 2000*; *Roulston & Goodell, 2011*), which can be problematic in agricultural landscapes because bee species foraging outside the restricted flowering period of pure culture dominated agricultural settings might be deprived of adequate alternative food sources. This phenomenon has been termed nutritional mismatch and has been suggested as one potential direct driver for the apparent decline in some bee populations (*Vaudo et al., 2015*).

In order to ease nutritional stress on bee populations in modern agricultural settings the establishment of alternative foraging habitats has been incentivized via agro-environmental management schemes in the EU and elsewhere (*Goulson et al., 2015*; *Lye et al., 2009*; *Phillips & Lowe, 2005*; *Potts et al., 2015*; *Vaughan & Skinner, 2008*). Such schemes seek to provide bees with flower resources outside the mass flowering periods of commercial crops in additional supplement to potential alternative resources (e.g., flowering plants in the field margins), but have traditionally focused solely on providing plants to attract and sustain social bees, particularly bumblebees (*Vaudo et al., 2015*; *Wood, Holland & Goulson, 2017*). Only recently has the important role of solitary bees in this context been recognized (*Scheper et al., 2015*). However, in all cases increasing the quantity of flowering plants and the lengthening timing of flowering alone is likely insufficient to maintain healthy bee populations. The quality of floral resources (including sugar concentration in nectar and protein content of pollen) and its natural variability also plays a major role in bee health with direct consequences, at least for the fitness of social bees (*Brodschneider & Crailsheim, 2010*; *Di Pasquale et al., 2013*; *Roulston, Cane & Buchmann, 2000*; *Tasei & Aupinel, 2008*; *Vaudo et al., 2016*; *Vaudo et al., 2015*). Such qualitative aspects of bee nutrition should be taken into consideration to develop a complementary and nutritionally optimized resource base for bee populations in agricultural landscapes (*Vaudo et al., 2015*).

To facilitate the integration of flower resource quality in pollinator management schemes we compiled a geographically explicit data base of pollen quality (measured as crude protein content) offered by bee-visited flowers in an agricultural setting. Given that pollen is the main protein source for bees' offspring, crude protein concentration in pollen is directly linked to the amount of protein bees can extract from their habitat and has traditionally served as a proxy for pollen quality (*Roulston, Cane & Buchmann, 2000*; *Roulston & Goodell, 2011*; *Vaudo et al., 2015*). However, most of the evidence supporting the importance of crude protein concentration for the fitness of bees (*Brodschneider & Crailsheim, 2010*) and their ability to adjust their individual (*Ruedenauer, Spaethe & Leonhardt, 2015*; *Ruedenauer et al., 2018*) or collective response according to their protein requirements (*Fewell & Bertram, 1999*; *Pernal & Currie, 2001*) stems from social bees. In the case of solitary bees the picture is less clear (*Eckhardt et al., 2014*; *Haider, Dorn & Müller,*
*2013*; *Nicholls & Hempel de Ibarra, 2017*; *Vanderplanck et al., 2014*), but in many cases solitary bees will also likely benefit from increased protein availability. Only in recent years additional factors, such as amino acid composition, lipids and potentially secondary plant metabolites, have emerged as variables potentially shaping pollen quality and consequently foraging decisions for bees in particular for solitary ones (*Cook et al., 2003*; *Nicholls & Hempel de Ibarra, 2017*; *Palmer-Young et al., 2019*; *Sedivy, Müller & Dorn, 2011*; *Vaudo et al., 2016*). In contrast to the extensive literature on pollen crude protein content (*Roulston, Cane & Buchmann, 2000*), data on these emerging quality factors are still scarce and often ambiguous and were consequently not included in this first analysis. However, it would be a logical and important next step to merge these data with information on secondary quality characteristics and bee ecology to utilize their full potential.

To give an overview of the collected data and potential applications we used the generated database to compare the crude protein concentration of pollen resources bees can encounter in agricultural landscapes in- (crop and weeds) and off-field (wild plants) around the globe. In a second step we identified crops, which provide bees with low, likely sub-optimal pollen and protein and suggest plants which could serve as high-quality alternative protein sources during times of need.

## MATERIALS AND METHODS

### Data collection and categorization

Data were collected, categorized and analyzed with minor modifications as previously described (*Pamminger et al., 2018*). Specifically, in 2018 we searched the literature for records on pollen quality in bee-visited flowers using ISI Web of Knowledge and Google Scholar. We used the search terms: flower AND pollen AND protein, adding either pollinator or bee as an additional term. Using these results, we identified relevant publications by reading the title and abstract. Based on this refined literature list we extended our search to the literature cited within these publications. In addition, we extended our data gathering efforts to the French and German literature to provide a more complete picture and make this information accessible to the English-speaking scientific community.

### Plant selection

Following (*Pamminger et al., 2018*) plant species were categorized as bee-visited if either bee visitation or pollination was directly observed or the flowers were explicitly classified as "melittophile" based on their floral characteristics by the study authors. In addition, we used information from the literature, the USDA pollinator manual (*Fowler, Rotheray & Goulson, 2016*; *McGregor, 1976*) and the expertise of BASF plant experts for cross-validation of the derived classifications.

### Geographic localization

Following (*Pamminger et al., 2018*) we chose to map the plant distribution on a continental scale because this information was available for the majority of plant species included in the data set. We decided to choose the Panama Canal as separation line between North and

South America, the Urals and the Black Sea to separate Europe from Asia and the Suez Canal to separate Asia and Africa. Using the Encyclopedia of Life (http://eol.org/) as the main source for plant distribution we recorded the presence and absence of collection records of each plant species on the five continents. This very broad geographical classification is intended as a first attempt to make this information geographically explicit and should serve as a starting point to add more detailed information on the local geographic (e.g., national or region) or habitat characteristics in the future. Such information will be vital to make more precise predictions about temporal quality dynamics in agricultural landscapes around the globe.

## Categorization of crop weed and wild plants

Following (*Pamminger et al., 2018*) the selected plants were categorized as crop species if they were listed as "cultivated crops" in governmental databases (e.g., USDA: https://plants.usda.gov and European commission plant variety catalogue: https://ec.europa.eu, *McGregor, 1976*), the open primary literature or were known as such to our BASF crop experts. All remaining plants without such records were categorized as non-cultivated. In a second step these non-cultivated plants were categorized either as a weed species, if they were listed in one of the following agricultural or governmental weed data resource (USA Noxious weed database, https://plants.usda.gov; Australia weeds, http://www.environment.gov.au; or industry compendium; *Bayer, 1992*), or as wild plants if no such record was found. This separation was chosen as wild plants which are able to invade managed fields will likely (1) possess different traits (e.g., physiology, resistance traits etc.) and (2) will experience improved habitat quality (e.g., nutrients and water). Both factors will likely influence resource quality and thus merit a separate category. Once a plant was categorized (crop, weed or wild) it was categorized as such in all regions where it was present. For genus level analysis, we kept the overall classification, but in cases were genera contained a mix of crop and wild species we refer to them as "crop containing genera".

## Pollen quality

We used crude protein content in pollen (dry mass) as proxy for pollen quality. Direct measurements of total nitrogen (N) content were also included and the corresponding crude protein content was calculated using a conversion factor of 6.25 (*Roulston, Cane & Buchmann, 2000*). This simple quality characteristic was chosen because it is the most frequently reported quantitative measurement of pollen quality in the literature and is directly linked to social bee fitness (*Brodschneider & Crailsheim, 2010*; *Hass et al., 2019*; *Vaudo et al., 2015*). Additionally, it is likely that some solitary bees will also benefit from increased protein supply. In addition to protein content, other quality criteria (e.g., amino acid composition, lipid content and micronutrient composition) are also important markers for resource quality (*Vaudo et al., 2015*). In the future, such additional quality markers would make a valuable addition to the database to get a more complete understanding of pollen quality. Whenever available, we report multiple measurements for individual plant species since resource quality is likely influenced by local conditions and these data can give important insights into the within-species variability of the trait in question.

## Analysis
### *Pollen quality and its variability in bee visited plants*
In the first part of the analysis we focused on the broad picture of pollen quality and its associated variation within a given plant community (crop, weeds and wild) on all relevant continents around the globe. As habitat (e.g., resource availability) will likely influence pollen quality we considered multiple measurements of the same species in different habitats as independent (i.e., used as the basis for the recorded sampling size). This approach was chosen as the aim for this paper was to present the overall natural observed variation and broad patterns in pollen crude protein around the globe. In case this data would be used to identify potential drivers for the observed variation a phylogenetically controlled approach would be called for.

### *Pollen quality offered by plant genera*
In a first step we compared the quality of crop genera in terms of pollen quality using all genera for which we had more than three observations. In a second step we compared these results to the quality patterns when using plant species (wild and weed) or variety (crop) mean as basis in order to investigate the potential influence of overrepresented plant species on the broad patterns. We categorized pollen as either crude protein content as either sufficient (above 20% crude protein content) or low quality (below 20% crude protein content), consistent with the traditional labeling of some plant species (e.g., Sunflower (*Helianthus annuus* L.) and Maize (*Zea mays* L.)) as providing low quality pollen for social bees (*Day et al., 1990*; *Maurizio & Grafl, 1980*; *Nicolson & Human, 2013*; *Pernal & Currie, 2000*; *Tasei & Aupinel, 2008*). This classification is not intended as a clear-cut criterion, but rather as a means to identify crop genera where nutritional intervention might provide the strongest benefit.

### *Statistics*
Both statistical analysis and figure generations were done in R v. 3.3.3. In order to describe the broad geographic and phylogenetic patterns of protein concentration in pollen, we used summary statistics. We report sampling size (number of measurements reported in the literature), mean, median and 10th and 25th percentile for the geographic patterns and a graphical analysis of the of both the geographic and genus level measurements.

## RESULTS

In total, we found 316 measurements of percent crude protein content, of which 302 could be unambiguously attributed to one plant category (see materials and methods) and used for analysis. Protein concentrations ranged from 10–61%, with the majority of these measurements from wild flowers ($N = 127$, crop $N = 94$, weed $N = 81$). In general, the data are evenly spread across regions and plant categories (Table 1), except only limited data were available for wild plants in Africa ($N = 14$). Overall, we find small differences in the median protein levels between communities with crops having slightly lower concentrations (median$_{crop}$ = 25.2%) than wild species (median$_{wild}$ = 28.5% see Fig. 1). When looking at the regional scale we find this variation reflected in the New

**Table 1** Summary statistic of the crude protein concentration [%] in pollen of crop, weed and wild plant communities across the globe.

| Region | Community | N | Median | Mean | 10th Percentile | 25th Percentile |
|---|---|---|---|---|---|---|
| Global | ALL | 302 | 26.7 | 29.1 | 16.3 | 21.8 |
| | Crop | 94 | 25.2 | 26.6 | 16 | 21.6 |
| | Weed | 81 | 27.1 | 29.3 | 16.2 | 21 |
| | Wild | 127 | 28.5 | 31.6 | 18.5 | 23.7 |
| Europe | ALL | 178 | 25.3 | 27 | 15.8 | 19.7 |
| | Crop | 82 | 25.1 | 25.8 | 15.7 | 21.6 |
| | Weed | 59 | 24.4 | 26.9 | 16.2 | 20 |
| | Wild | 37 | 28.2 | 31.5 | 18.1 | 21.9 |
| North America | ALL | 220 | 28.5 | 31.1 | 16.6 | 22.9 |
| | Crop | 83 | 25.6 | 26.9 | 15.7 | 21.7 |
| | Weed | 67 | 28.2 | 30.6 | 16.6 | 21.8 |
| | Wild | 70 | 38.3 | 36.4 | 21.8 | 28 |
| South America | ALL | 153 | 27.1 | 29.6 | 16.2 | 21.9 |
| | Crop | 75 | 25.6 | 26.7 | 15.6 | 21.7 |
| | Weed | 49 | 24.9 | 28.8 | 16.2 | 19.2 |
| | Wild | 29 | 40.4 | 38.5 | 26.6 | 28.6 |
| Africa | ALL | 130 | 25.8 | 27.2 | 15.7 | 19.2 |
| | Crop | 66 | 25.1 | 26.1 | 15.3 | 19.9 |
| | Weed | 50 | 25.8 | 28.2 | 16.2 | 19.2 |
| | Wild | 14 | 28.1 | 28.5 | 16.2 | 20 |
| Asia | ALL | 150 | 25.8 | 27.4 | 15.9 | 20.1 |
| | Crop | 76 | 25.2 | 25.9 | 15.7 | 21.2 |
| | Weed | 48 | 24.7 | 27.5 | 16.2 | 19 |
| | Wild | 26 | 28.2 | 31.2 | 18.5 | 23.8 |
| Australia | ALL | 200 | 24.9 | 26.4 | 16.2 | 21.1 |
| | Crop | 76 | 24.6 | 25.8 | 15.7 | 21 |
| | Weed | 57 | 23.9 | 26.6 | 16.1 | 18.3 |
| | Wild | 67 | 25.2 | 27 | 18.8 | 22.8 |

World communities (North America and South America see Fig. 1). In all other regions such variation seems less pronounced or absent. When looking at genus level differences based on all observations we find pronounced variation between genera (Fig. 2), with two genera providing particularly high-quality pollen (Solanum and Senna) and two genera representing globally important crops (*Zea* and *Helianthus*) offering pollen with low protein content (Fig. 2). When comparing these results with the pattern based on species/variety means we find only minor deviations between the two analysis and no change in the overall genus level patterns (Table S1).

## DISCUSSION

Overall crude pollen protein content in bee-visited plants is around 26% with similar values for all categories (Fig. 1, Table 1). The only apparent deviation from these values

 

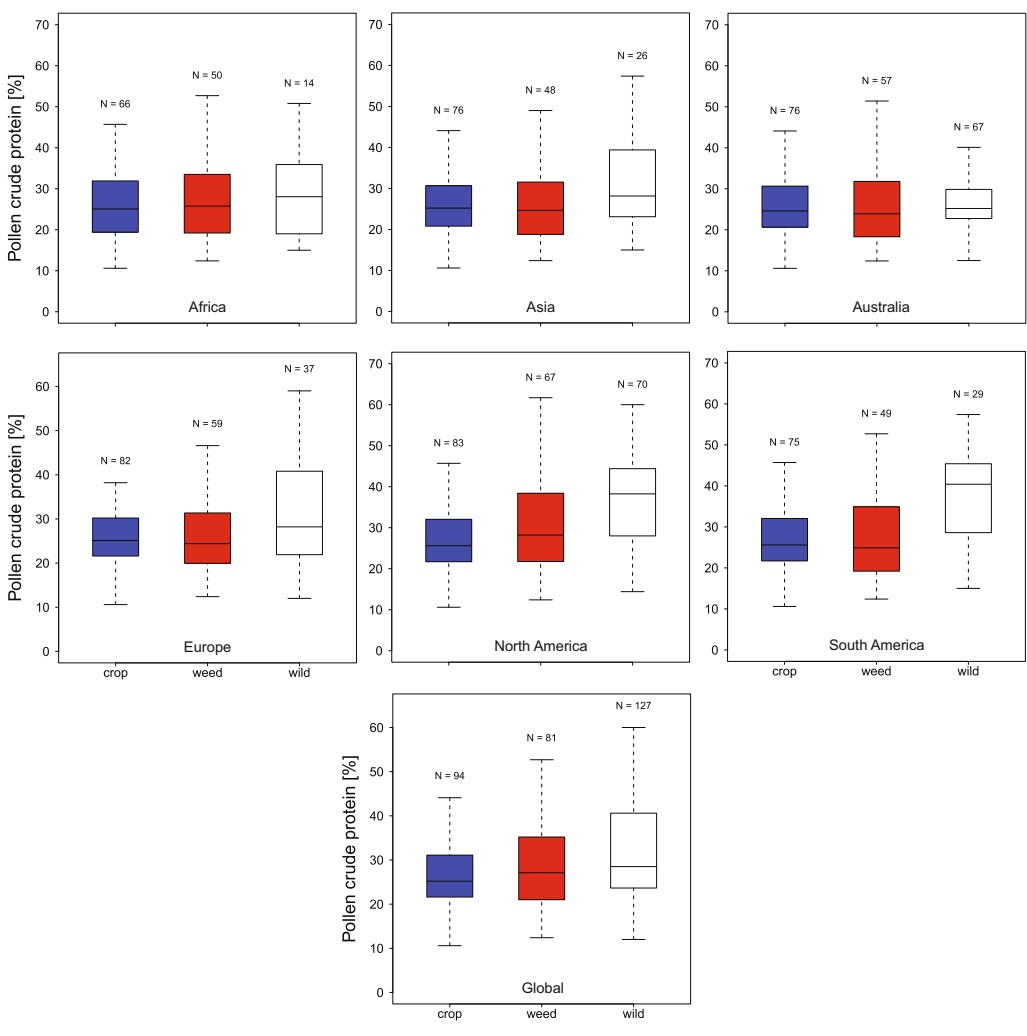

**Figure 1** **Pollen Quality of bee visted flowers in plant communities around the globe.** Summarizes the total crude protein concentration in percent in agricultural landscapes on a continental as well as global basis. We present data for Africa, Asia, Australia, Europe, North America, South America and Global for crop (blue), weed (white) and wild plant (red) communities. Sampling size (N) refers to total measurements of protein concentrations recorded in the literature for a given category.

appear in the wild plant communities of the New World (median North America = 38.3% and median South America = 40.4%), while plant communities in the other regions have comparable pollen protein concentrations (Fig. 1, Table 1). We found that only six genera offer pollen with low protein concentration (crude protein < 20%), including two main crop species: sunflowers (*Helianthus:* median around 15%) and maize (*Zea:* median around 16%). Most other plant genera offer pollen of comparable protein concentration (Fig. 2).

Interestingly, protein content in wild plants of the new world appear somewhat elevated while crop and weed species exhibit similar protein levels worldwide. In the latter case this is expected as these plant communities are more homogenous in all regions as a result of their intended (crop) or involuntary introduction (weeds) around the globe. In contrast,

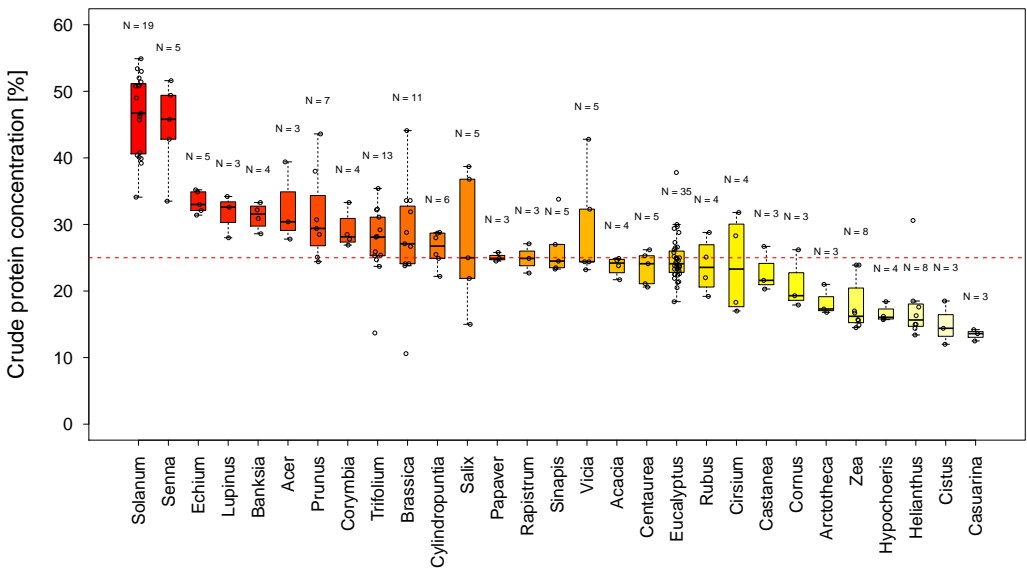

**Figure 2 Pollen quality of bee visited plant genera in agricultural landscapes.** Shows the distribution of crude protein concentration in percent among all genera for which more than three measurements were available. Genera below the red line offer pollen of low protein concentration. Sampling size (N) refers to total measurements of protein concentrations recorded in the literature for a given genus (see Table S1 for the number of species and varieties per genus).

most wild plants in this study are geographically restricted (non-global distribution), suggesting that this potential pattern could be caused by differences in the community composition between regions. However, caution needs to be taken as this pattern could also be the result of incomplete sampling (e.g., in Africa) or could represent sampling bias because the results are based on peer-reviewed publications and could simply reflect the focus of the study authors.

Two genera (*Solanum* and *Senna*) were identified with higher pollen protein concentration than the majority of plant genera visited by bees (Fig. 2). Interestingly both genera possess poricidal anthers, which naturally restrict pollen access by non-specialized pollinators and potential pollen robbers (*De Luca & Vallejo-Marín, 2013*). The resulting reduced risk of pollen loss might have contributed to the evolution of increased pollen quality. Most genera have similar pollen protein concentrations, which are at a level likely suitable to support bee populations (*Day et al., 1990*; *Di Pasquale et al., 2013*; *Roulston, Cane & Buchmann, 2000*; *Tasei & Aupinel, 2008*; *Wcislo & Cane, 1996*). In addition, our findings support the conclusion that maize and sunflower have low pollen quality in terms of protein content (*Hass et al., 2019*; *Maurizio & Grafl, 1980*). While sunflower can offer high quality nectar as an alternative reward (*Maurizio & Grafl, 1980*; *Tepedino & Parker, 1982*) to attract bees, maize is primarily wind pollinated and does not offer nectar rewards. Therefore, sunflower, and in particular maize, is considered less attractive for bees and are likely only occasionally visited in the absence of alternative pollen sources (*McGregor, 1976*). These results suggest that it might be of particular interest to supply

bees in sunflower- and maize-dominant agricultural landscapes with high quality pollen species, adjusted for the season and region (*Hass et al., 2019*).

Given that pollen is the primary protein source for the majority of bees it is important to ensure adequate protein supply when planning alternative foraging areas (*Vaudo et al., 2015*). While it is clear that increased protein supply can be beneficial to developing larvae and insufficient protein supply can result in larval malnutrition with clear adverse effects (*Brodschneider & Crailsheim, 2010*; *Di Pasquale et al., 2013*; *Hass et al., 2019*; *Tasei & Aupinel, 2008*), there is only limited support for a simple relationship between crude protein concentration and bee fitness (*Babendreier et al., 2004*; *Brodschneider & Crailsheim, 2010*; *Di Pasquale et al., 2013*; *Tasei & Aupinel, 2008*). It is likely that in addition to protein content, other quality markers such as lipid content, amino acid composition and secondary plant metabolites might play an important role in determining pollen quality for bees (*Day et al., 1990*; *Hass et al., 2019*; *Maurizio & Grafl, 1980*; *Nicholls & Hempel de Ibarra, 2017*; *Nicolson & Human, 2013*; *Pernal & Currie, 2000*; *Ruedenauer et al., 2018*; *Tasei & Aupinel, 2008*). In contrast to nectar quality (sugar concentration), there is only some evidence that bees can reliably separate high from low quality pollen and adjust their collecting behavior according to their needs (*Nicholls & Hempel de Ibarra, 2017*). However, recent work suggest that bumblebees are able to do this impressive feat on an individual level (*Ruedenauer, Spaethe & Leonhardt, 2015*) and that honeybees can on a collective level (likely using feedback from their larvae; *Pernal & Currie, 2001*; *Ruedenauer et al., 2018*) if quality differences are sufficiently large. These promising findings suggest that by adding selected high quality pollen resources to agricultural landscapes social bees would likely benefit directly, while at least some solitary bees will likely be able to utilize such additional resources (*Hass et al., 2019*; *Scheper et al., 2015*).

## CONCLUSIONS

This paper is a first step to collect the available data on pollen quality, in terms of protein content, offered to bee visited flowers in agricultural habitats and make them easily accessible in an electronic format. In the future this dataset could be combined with more detailed information on both pollen quality (e g. lipids, amino acid composition and secondary plant metabolites) as well as plant traits (e.g., flowering period and local geographic distribution), which could enable improved bee management practices and potentially more realistic landscape level modelling approaches to facilitate bee conservation in modern agricultural habitats.

## ACKNOWLEDGEMENTS

We like to thank Dr. Adric Olson for his critical reading of an earlier draft of the manuscript.

### Funding

The publication cost were covered by ECPA - the European Crop Protection Association. The funders had no role in study design, data collection and analysis, decision to publish, or preparation of the manuscript.

### Grant Disclosures

The following grant information was disclosed by the authors:
ECPA - the European Crop Protection Association.

### Competing Interests

All authors are employed by BASF (an agricultural solution provider).

### Author Contributions

- Tobias Pamminger conceived and designed the experiments, performed the experiments, analyzed the data, contributed reagents/materials/analysis tools, prepared figures and/or tables, authored or reviewed drafts of the paper, approved the final draft.
- Roland Becker contributed reagents/materials/analysis tools, authored or reviewed drafts of the paper, approved the final draft, contributed ideas, interpreted data.
- Sophie Himmelreich, Christof W. Schneider and Matthias Bergtold authored or reviewed drafts of the paper, approved the final draft, contributed ideas, interpreted data.

### Data Availability

The following information was supplied regarding data availability: Pamminger, Tobias (2019): 2018_bee_protein_concentration_PeerJ_R1. figshare. Dataset. https://doi.org/10.6084/m9.figshare.8174627.v1.

### Supplemental Information

Supplemental information for this article can be found online at http://dx.doi.org/10.7717/peerj.7394#supplemental-information.

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
