# Peer review of "Pollen report: quantitative review of pollen crude protein concentrations offered by bee pollinated flowers in agricultural and non-agricultural landscapes"

_PeerJ, doi:10.7717/peerj.7394_

## Round 0.1 · original submission · Major Revisions

Thank you for your submission. Please address the reviewers comments, particularly with respect to the statistical analyses. I look forward to receiving your revised submission.

Reviewer 1 ·

Basic reporting

The manuscript is clearly written and structured and it is easy to follow the authors’ rationale and findings. I have suggested some additional references which might be included in my general comments below. I had some trouble with the formatting of the raw data file, so it wasn't possible to examine this in much detail.

Experimental design

The database is a useful resource and though, as you acknowledge, there are shortcomings in using only crude protein content as a measure of nutritional quality, the data will be of assistance in planning nutritional studies, agri-env schemes etc. and hopefully can be built up on in future if data for other macronutrients becomes more widely available.

In the analysis did you use raw data from the cited papers, or the averaged values for each species?

The rationale for comparing wild versus cultivated plants is fairly obvious, but the comparison of wild flowers versus ‘weeds’ needs more justification, since in reality there is overlap in these categories.

Validity of the findings

I question why you have chosen to publish the nectar and pollen results separately. Both resources are important for bee nutrition, and for those plant species where you have data on both matrices it would be interesting to compare and calculate the overall nutritional quality of a species. In terms of provisioning bees in agricultural landscapes, while some plants may provide a poor pollen source, this can be compensated by a good nectar supply or visa versa, so when making recommendations as you do in your discussion, it would be best to consider both floral rewards where possible, and I feel this would make your conclusions stronger and more informative.

Additional comments

Introduction:
Line 41: primarily
Line 58: Not all bee species will visit a particular crop, so weeds/wild flowers are (in theory) also providing alternative, more appropriate/attractive foraging sources during crop flowering, as well as extending the flowering period.
Line 60: Consider also citing Wood et al. (2017) J. Applied Ecology
Line 63: You should probably explicitly mention that pollen protein content is highly variable between species e.g. Roulston & Cane (2000) Eco Monographs
Line 82: See also following papers, among others, on solitary bee nutrition; Vanderplanck et al. (2014) PLoS ONE; Haider et al. (2013) Fun. Ecology; Eckhardt et al. (2014) J. Animal Ecology
Line 84: Lipids seem to be important, as well as the ratio of protein to lipids e.g. Vaudo et al. (2016) PNAS, JEB
Methods:
Line 115- do you mean strictly if pollination has been observed or if pollination and/or visitation was observed?
You could also include data from Fowler et al. (2016) Insect Conservation & Diversity
Results:
Figure 1- clarify in the legend what N refers to- presumably species, but would be good to include the number of genera too in each category as there is likely some co-variation in protein content
Figure 2: Why are some genera listed in red?
Discussion:
Interesting that Senna and Solanum have highest pollen protein concentration- both have poricidal anthers to restrict collection/wastage of pollen by bees, which might be worth mentioning.
Line 261- True that they can discriminate, but seemingly only when differences in quality are sufficiently large (>30% difference in pollen content). However your results suggest that differences in crude protein content within and between most genera were not very large.
Line 263- This is a bit vague/over-simplified- increased how?

Reviewer 2 ·

Basic reporting

This manuscript is well written, and the introduction frames the experiment well. I have suggested the addition of raw data to all of the plots and have provided an example of such a plot. I have detailed more explicit comments regarding the plots in my general comments section. All of the data are provided but need minor revision to make them more easily accessible to a broad audience (detailed in general comments).

Experimental design

At its core this is a valid compilation and re-analysis of an interesting global dataset of pollen protein contents. Provided that the authors can address the statistical issues outlined in the general comments section, any findings supported by those corrected analyses will be sound.

Validity of the findings

Despite the strengths detailed above, I believe that in their current state the statistical analyses need revision and are not sufficient to support the authors’ conclusions. I have detailed the issues in the general comments section of my review.

Additional comments

This manuscript provides a valuable analysis of pollen protein content in agricultural and wild landscapes. The authors gathered data from eight previous publications on the protein contents of crop, weed, and wild plants and use these data to conclude that wild plants produce higher protein pollen than crop or weed plants on a global scale. This pattern is largely driven by plants in the new world, but there is a non-significant trend towards a similar pattern in Europe.

Though these findings are potentially very interesting, there are some serious issues with the statistical analyses that need to be fixed before the authors conclusions are fully supported. I have organized my comments into Major Issues that I feel need to be addressed in order for the data, analyses, and conclusions to be sound and Minor Comments that deal primarily with the presentation and communication of the findings. Further I have directly addressed the PeerJ editorial criteria at the end of this review.

Major Issues

• I am very concerned about the authors’ analytical methods regarding whether or not crops, weeds, or wild plants have higher pollen protein content. Based on the sample sizes presented in figure 1, it appears that they are treating each observation in their data as independent despite ~85 of their observations being within species replicates. It is not clear in the description of their statistical models (lines 183-189) whether or not the authors accounted for this in their models, but figure 1 suggests they did not. If the authors did not account for this repeated measurement, they need to. Otherwise if they did somehow account for the non-independence of repeated measures on a single species they need to describe how they did this.

• Similarly there are two issues with the authors analysis of genus level variation. First, at line 173, the authors say they only included genera that had more than three species level records in this analysis, but they did not do this. They have included some genera that have three or more repeated measure of a single species but not three different species in the genus (e.g. Lupinus, Zea, Hypochoeris). Secondly, these data seem to call for a phylogenetically corrected model, and the authors should attempt to apply some sort of phylogenetic correction or explain why it would be inappropriate to do so.

• The authors use two schema for categorizing plants in their dataset: 1) domestication and weediness status, and 2) genus. At various points in the manuscript they refer to genera of plants as either crop, wild, or weed (e.g. line 207), but not all members of every genus in the dataset are entirely agricultural or wild. This is especially true for members of Solanum which the authors refer to as a “crop genus” despite only 4 of the 18 species in the authors’ database being crop plants. The authors should be careful regarding this point throughout the manuscript.


Minor Comments

• L64-67: The authors should move the citations after the final word of this sentence to increase readability.

• L76 and elsewhere: Throughout the work there is an issue with the Roulston, Cane, and Buchmann 2000 citation. First, T.Roulston’s name is spelled incorrectly in the supplemental data file (currently written as Roulstone in the, but the correct surname is Roulston). Secondly in all of the in-text citations the authors have used T.Roulston’s first name instead of his surname (e.g. line 76 should read “Roulston, Cane et al. 2000”).

• L84: The authors should consider including lipid content here (as they do at line 158 and 253).

• L279-280: There is a missing line here, and I believe the authors may have inadvertently deleted some text.

• Works Cited: as written the works cited section is extremely difficult to read, and its formatting should be revised for clarity.

• Figures 1 and 2: The authors should add raw data as points overlaid over these box plots (to provide readers with a better sense of the distribution of the data) as long as it does not make the plots overly complicated. As an example, I have produced the an example figure (an attached version of figure 2 that both corrects for the issues mentioned above regarding the inclusion of genera that do not have 3 species, and which also includes raw data). I will include the code to generate this plot with my review.

• Figure 2: It appears that the authors have color coded the genera, but the logic behind this color coding is not clear in the figure description. Perhaps red genera are crops, and black genera are wild plants. If this is the case the authors should consider how they might color code genera that contain crops, weeds, and wild plants (e.g. Solanum).

• Supplementary data file: The data are uploaded as a comma separated values file (.csv), but are semi-colon delimited. This may cause issues for some readers who open the file in common spreadsheet software (e.g. Microsoft Excel). The authors should either change the file type to .txt or change the delimiter to a coma.

Annotated reviews are not available for download in order to protect the identity of reviewers who chose to remain anonymous.

---

## Round 0.2 · Minor Revisions

Please make the remaining revisions suggested by reviewer 2 and re-submit. Thank you!

Reviewer 1 ·

Basic reporting

No comment

Experimental design

No comment

Validity of the findings

No comment

Additional comments

The authors have either addressed or provided satisfactory justification for all of the comments and queries I raised in response to the original version of the manuscript. The paper will be a very informative and useful addition to the field of pollination ecology.

Reviewer 2 ·

Basic reporting

Overall the reporting is clear and professional. I only have suggestions on two minor corrections, and the inclusion of one extremely pertinent citation.

Comments:

L63-65: This sentence as written is not 100% clear. Consider revising, “However, in all cases INCREASING the quantity of flowering plants, and LENGTHENING the timing of flower alone are likely…”

L93: It would be worth citing the recent large study of pollen secondary metabolites – Palmer-Young et al. (2018) Chemistry of floral rewards: intra: and interspecific variability of nectar and pollen secondary metabolites across taxa. Ecological Monographs 89(1). E01335.

L232: “a” should be “and”

Experimental design

Though the authors have decided to only report summary statistics, I still feel that treating each observation as independent is fundamentally flawed. I have detailed my reasoning, suggested a solution, and have detailed specific lines in the manuscript where there are issues. Further, the authors need to report the number of species represent in each genus (and report the methods they actually used for generating figure 2 - see comment below).

Comments:

L178-180: Though the authors have chosen to not use any statistical hypothesis testing, there is still an issue with treating multiple observations of the same plant species as independent. Is there evidence that within species variation (due to the factors described in the text) is higher than among species variation? There is no issue reporting multiple values for a single plant, but there is an issue treating each observation as independent as it clearly is not.

Instead of treating each measurement from a species as an independent observation when calculating summary statistics, the authors should consider calculating the mean of each species first, then calculate the genus means based on each species level mean. This is a major issue if any given species differs from the true genus level mean and is overrepresented in the sample. In this case the genus level measurement will always be biased in the direction of the more sampled group. I have done some simple simulations (PDF of some code and figures attached) that demonstrate the issue here.

L187-188: The authors say here that they only used genera that had observations for at least 3 species, but this is pretty clearly still not the case in figure 2. For example, they have 5 observations for Echium, but these are only from two species and 3 observations for Lupinus but from only one species. If the authors are going to include the genera with less than three species represented they should not only report the number of observations for each genus, but also the number of species represented in each genus (at line 201, Table 1, and/or Figure 2).

Validity of the findings

I still feel these are extremely useful data, and the authors should be lauded for compiling this data base. That said there are still a few issues with their interpretation of these results (see my comments above re: species level replication).

Additional comments

Overall the manuscript has improved after revision, and needs only very minor revision before it is acceptable for publication in PeerJ.

Annotated reviews are not available for download in order to protect the identity of reviewers who chose to remain anonymous.

---

## Round 0.3 · accepted · Accept

Thank you for addressing the reviewer comments to the best of your ability. Congratulations.